# Clustering Analysis Identified Three Long COVID Phenotypes and Their Association with General Health Status and Working Ability

**DOI:** 10.3390/jcm12113617

**Published:** 2023-05-23

**Authors:** Marta A. Kisiel, Seika Lee, Sara Malmquist, Oliver Rykatkin, Sebastian Holgert, Helena Janols, Christer Janson, Xingwu Zhou

**Affiliations:** 1Department of Medical Sciences, Occupational and Environmental Medicine, Uppsala University, 751 85 Uppsala, Sweden; sebastianholgert@hotmail.com; 2Department of Neurobiology, Care Sciences and Society, Primary Care Medicine, Karolinska Institute, 171 77 Stockholm, Sweden; seika.lee@ki.se; 3Department of Statistics, Uppsala University, 751 20 Uppsala, Sweden; sara.malmquist.4387@student.uu.se (S.M.); oliver.rykatkin.8816@student.uu.se (O.R.); 4Department of Medical Sciences, Infection Disease, Uppsala University, 751 85 Uppsala, Sweden; helena_janols@yahoo.se; 5Department of Medical Sciences: Respiratory, Allergy and Sleep Research, Uppsala University, 751 85 Uppsala, Sweden; christer.janson@medsci.uu.se; 6Department of Medical Sciences: Clinical Physiology, Uppsala University, 751 85 Uppsala, Sweden

**Keywords:** long COVID phenotypes, persistent long-term symptoms, COVID-19, working ability, general health status

## Abstract

Background/aim: This study aimed to distinguish different phenotypes of long COVID through the post-COVID syndrome (PCS) score based on long-term persistent symptoms following COVID-19 and evaluate whether these symptoms affect general health and work ability. In addition, the study identified predictors for severe long COVID. Method: This cluster analysis included cross-sectional data from three cohorts of patients after COVID-19: non-hospitalized (n = 401), hospitalized (n = 98) and those enrolled at the post-COVID outpatient’s clinic (n = 85). All the subjects responded to the survey on persistent long-term symptoms and sociodemographic and clinical factors. K-Means cluster analysis and ordinal logistic regression were used to create PCS scores that were used to distinguish patients’ phenotypes. Results: 506 patients with complete data on persistent symptoms were divided into three distinct phenotypes: none/mild (59%), moderate (22%) and severe (19%). The patients with severe phenotype, with the predominating symptoms were fatigue, cognitive impairment and depression, had the most reduced general health status and work ability. Smoking, snuff, body mass index (BMI), diabetes, chronic pain and symptom severity at COVID-19 onset were factors predicting severe phenotype. Conclusion: This study suggested three phenotypes of long COVID, where the most severe was associated with the highest impact on general health status and working ability. This knowledge on long COVID phenotypes could be used by clinicians to support their medical decisions regarding prioritizing and more detailed follow-up of some patient groups.

## 1. Introduction

A proportion of patients infected with the severe acute respiratory syndrome coronavirus-2 (SARS-CoV-2) experience long-term persistent symptoms following coronavirus disease 2019 (COVID-19). These symptoms are referred to as long COVID [1]. In accordance with the World Health Organization’s (WHO) case definition, long COVID is a condition usually occurring three months following the onset of confirmed or probable COVID-19, with symptoms that last at least two months and cannot be explained by other diagnoses [2,3]. Long COVID may occur in patients after mild infection and in those that required hospitalization [4,5]. The true prevalence of long COVID symptoms remain unknown and varies across the international studies [6,7]. In accordance to WHO, there might be 17 million people who have suffered long COVID in Europe [7]. The clinical presentation of long COVID is highly variable and heterogenous, and over 50 symptoms have been linked to long COVID [8]. Fatigue, cognitive problems and dyspnea are the most commonly reported symptoms [9]. Female sex, higher age, low socioeconomic status, pre-existing conditions including cardiovascular disease and diabetes mellitus type 2 and a high body mass index have been indicated as risk factors for long COVID. However, whether those factors are associated with specific symptoms is not clear [10].

So far, there are no effective treatments for long COVID. Furthermore, the diagnostic criteria and clinical spectrum of this condition are not fully understood. Early identification of long COVID patients with poorer recovery and higher symptom load could lead to more individual symptomatic rehabilitation for these with poor outcomes [11]. Existing research suggested that the use of machine learning method such as clustering algorithm may distinguish patients with different severity of long COVID [12,13,14]. Some attempts have been made to distinguish cluster of patients according to persistent long-term symptoms associated with long COVID [13,14,15]. Baher et al. performed a cluster analysis on German patients after COVID-19 of different severity using a long-term persistent symptom scoring system called the post-COVID syndrome (PCS) score [15]. A higher PCS score was equal to the long COVID phenotype of higher severity that was linked to a lower quality of life [15]. A few previous studies of long COVID clusters have addressed work ability. Those reports showed that individuals with more severe post-COVID symptoms experienced disruption of work and productivity loss [14,16]. In addition, there are no previous studies on long COVID clusters in patients from Sweden. In order to fill these gaps, in this study we aimed to identify phenotypes of long COVID based on the PCS score to determine whether self-reported work ability and general health status is associated with the higher load of persistent symptoms. We also wanted to examine predictors of severe long COVID in Swedish previously non-hospitalized and hospitalized COVID-19 patients.

## 2. Materials and Methods

This study is part of the longitudinal project called “COMBAT post COVID” that aimed to investigate long-term health after COVID-19 and its consequences [17,18]. The project was conducted at Uppsala University Hospital, Sweden. This current study included three cohorts of adults (≥18 years old). The flow chart of the study population is shown in Figure 1.

The studies were approved by the Swedish Ethical Review Authority (2020-05707, 2021-01891, 2022-01261-01) and conducted in accordance with the Helsinki Declaration. All the included study subjects gave informal consent to use their survey answers. The founding sources had no role in study design, data collection and data analysis, data interpretation or writing of the report.

The cohort called “Non-hospitalized COVID” included subjects who underwent non-hospitalized/mild COVID-19. The enrolled subjects were symptomatic at onset and had a positive polymerase chain reaction (PCR) for SARS-CoV-2 in nasopharyngeal swabs between March and December 2020. The study population included mainly employees of Region Uppsala in particularly health care and resident care workers, since those groups were prioritized for early testing in the pandemic. Twelve months after infection, the patients answered a survey on their symptoms at onset and comorbidities. The survey was sent to 725 subjects 45–51 weeks after the COVID-19 diagnosis, by e-mail [19], or if no e-mail address was available, to the home address by ordinary post. In total, 401 answered the survey (response frequency 55%). This population has been described elsewhere [17,18].

The cohort called “Hospitalized COVID” included subjects hospitalized for COVID-19 at the Department of Infectious Diseases at Uppsala University Hospital. All the subjects had a positive PCR for SARS-CoV-2 and were registered in the local register of patients hospitalized due to the COVID-19 infection. The survey was done 12 months after infection. The cohort was collected at two different time points. Between April and June 2020, 99 individuals received a survey to their home address and 46 (46%) responded. Between April and May 2021, of 140 subjects hospitalized, 52 subjects (37%) participated in a COVID-19 follow-up visit at the Department of Respiratory, Allergy and Sleep Research, Uppsala University Hospital. These patients responded to the survey in addition to a physical evaluation.

The cohort called “post-COVID” included subjects enrolled to the post-COVID outpatient’s clinic in Uppsala. These patients were referred from primary care centers in Region Uppsala with remaining symptoms after at least 12 weeks of a microbiologically verified or probable COVID-19 infection (according to WHO definition, Delphi study) [3]. All these patients had fatigue/cognitive problems that significantly reduced their work capacity. All enrolled patients until May 2021 (n = 161) received the survey to their home address between October 2021 and May 2022 with a return envelope. Of those invited, 85 (53%) responded. Of the study subjects in the post-COVID cohort, 57 patients had a laboratory-confirmed COVID-19 infection, and 21 patients were hospitalized due to COVID-19 infection. The median follow-up time after the infection onset was 22 months.

The study subjects answered question (Appendix B) on whether they experienced one or several of 17 long-term persistent symptoms including dyspnea, cough, sore throat, nasal congestion, impaired taste and/or smell, heart palpitation, chest pain, vertigo, headache, muscle/joint pain, fatigue, memory/concentration problems, sleeping problems, depressive mood, anxiety, gastrointestinal problems (including nausea, vomiting and stomach pain) and skin problems.

The survey included questions regarding sociodemographic factors including education level, divided into university education (at least 3 years) or vocational collage (at least 2 years not equivalent to an academic degree) or upper secondary school; marital status (divided into married, living with partner, single, divorced and widow/-er); country of birth included Sweden and countries other than Sweden; work status including current work, studies, parental leave, sick leave, unemployment or retirement; smoking status; use of snuff; and weight and height that were used to calculate the body mass index (BMI) kg/m^2^. 

The participants were asked about the self-reported severity of symptoms at onset including very mild/mild, moderate, severe or very severe; as well as other disease diagnosed by a doctor including hypertension, heart disease (such as heart failure or acute myocardial infarction), hypo-/hyperthyroidism, diabetes mellitus type 2, lung disease (including asthma and chronic obstructive pulmonary disease), conditions requiring immunosuppressive treatment, cancer, depression/anxiety and chronic pain. These diagnoses were selected as they were considered risk factors for severe COVID-19 infection and the symptoms were the most common symptoms at infection onset according to the WHO/International Soil Reference and Information Centre (WHO/ISARIC) platform [20]. Age and legal sex (females or males) were determined based by the Swedish personal identification number.

General health status after the infection (at the time of answering the survey) was assessed by a modified version of the EQ visual analog scale (VAS) [20]. The numerical scale ranged from 0 = “worst imaginable health status” to 100 = “best imaginable health status”. The participants were also asked to retrospectively assess their health status before the COVID-19 infection study subjects using the same kind of VAS scale as above [21]. Work ability before and after the COVID-19 infection was assessed by a numerical scale ranging from 0 = “worst imaginable working ability” to 10 = “best imaginable working ability” [17].

All statistical analysis was performed by using R (R Core Team, 2023) [22]. For the descriptive statistics, the categorical variables are presented as proportions, continuous variables are presented as means and standard deviations and some skewed variables are presented with median and interquartile range (IQR). The differences between characteristic among the three cohorts were assessed by using analysis of variance (ANOVA), Kruskal–Wallis tests, Chi-square tests or Fisher’s exact test. The *p*-values for multiple comparisons were adjusted by the Benjamini and Hochberg (1995) method to control for the false discovery rate (FDR) [23].

The study subjects were divided into different groups by the K-Means clustering method. The ordinal logistic regression models were used for developing the PCS scores and selecting the final predicting model. The area under the curve (AUC) of the multiple classes was used to calculate the overlap with the K-Means clustering and the PCS clustering. A backward selection procedure was used for the optimal variable selection. The Pearson correlation coefficients were calculated for assessing the association between the clinical functions and the predicted PCS scores. The confidence intervals were calculated by the bootstrap methods for the correspondingly statistics, for these correlation coefficients and also for the AUC of the multiple class receiver operating characteristic (ROC) curve [24]. The flow chart and steps of the data analysis are shown in Appendix A. 

Sensitivity analysis was carried out in two ways. One kind of sensitivity analysis was done by randomly splitting the total cohort into the training set with 80% of the individuals and the test set with 20% of the individuals.

## 3. Results

### 3.1. Study Population Characteristics

In total, 584 subjects (response frequency 52%) from three sub-cohorts completed the survey and were included in this study. The non-hospitalized cohort had the lowest mean age, highest proportion of women and more subjects with higher education, as seen in Table 1. The hospitalized cohort had the highest mean age and predominantly consisted of men; they had highest BMI and the highest proportion of hypertension and diabetes mellitus type 2. The post-COVID cohort had the highest proportion of current smokers and the highest proportion of coexisting depression, anxiety and chronic pain. This group had the highest proportion of participants on sick leave and the highest reduction of work ability.

### 3.2. Cluster Analysis of Three Combined Sub-Population

Complete data on all 17 long-term persistent symptoms were available for 506 of the study subjects. The proportion of missing data on each symptom in the three included cohorts is shown in Appendix A. The Pearson correlation coefficient between the adjacent cluster number-specific PCS scores, i.e., cor (PCS__K_, PCS__(K+1)_), was r = 0.980 (*p* < 0.001) for K = 2, r = 0.992 (*p* < 0.001) for K = 3 and r = 0.989 (*p* < 0.001) for K = 4 (Appendix A). Therefore, we considered K = 3 as an optimal correlation coefficient. Then, an ordinal logistic regression was run for the ordered clusters using the 17 binary persistent symptoms as independent variables, as seen in Table 2. Each variable was assigned a weight according to the estimated model. Using the assigned weight, each individual then received a PCS score. The boxplots of the PCS scores for these three clusters are shown in Appendix A. Using the PCS scores, the study subjects were dived into three groups defined as none/mild phenotype (PCS score ≤ 13), moderate phenotype (PCS score > 13 and ≤40) and severe phenotype (PCS score > 40). A ROC curve confirmed the overlap between the K-Means clusters and the PCS score groups. The calculated multiple class AUC was 0.981 with a bootstrapped confidence interval (0.970, 0.990), indicating a good overlap between the K-Means clusters and the PCS score groups. The distribution of the PCS scores for each cohort are shown in Appendix A, and the principal component of the 3 K-Means clusters are shown in Appendix A. The K-Means clusters are ordered according to the summary of the three cluster centers. 

The majority of the study subject from the non-hospitalized cohort were included in the none/mild group, and the majority of the post-COVID cohort were included in the severe group (Figure 2).

### 3.3. Characteristic of the Phenotypes

Significant inter-class differences were shown between the three PCS score groups/phenotypes regarding several sociodemographic, clinical and functional factors (Table 3). The non/mild phenotype (PCS score ≤ 13) included the subjects with no persistent symptoms (PCS score = 0, n = 195) and less persistent symptoms (PCS score 1–13, n = 103). This group comprised more than a half of the whole study population. The non/mild phenotype group comprised the study subjects that were younger, more often never smokers but using snuff, more often currently working, with higher education, with the lowest BMI, with a low proportion of coexisting disease and were less often hospitalized due to COVID-19. The subjects within the severe phenotype were often on sick leave, more often current smokers, with the highest BMI and with the highest proportion of comorbidity. They were followed up after longer period of time from the acute COVID-19 infection but reported the largest number of persistent symptoms and the highest reduction of health status and working ability. The severe phenotype had a slightly higher proportion of hospitalized patients than the moderate phenotype.

### 3.4. The Relevance of Persistent Symptoms

There were 182 (45%) of subjects in the non-hospitalized cohort and 13 (13%) in the hospitalized cohort that did not report any persistent long-term symptoms, while all the subjects in the post-COVID cohort reported at least one persistent long-term symptom. The most common persistent symptom among subject of the three cohorts was fatigue, as shown in Figure 3A. Fatigue was also the most commonly reported persistent symptom in the moderate and severe phenotype, as shown in Figure 3B. The subjects with the severe phenotype group also often complained of cognitive impairment and depression, whereas the subjects from moderate phenotype often reported cognitive problems and dyspnea. The most commonly reported symptoms in the non/mild phenotype were impaired smell and taste. 

### 3.5. Cluster Analysis in the Test Population

The cohorts were randomly split into the training set (n = 407, 80%) and the test set (n = 99, 20%). Using the similar procedures as for the full dataset, three clusters were determined according to the PCS score correlations (Appendix A). Then PCS scores were cut into PCS groups by 12 and 40. The overlap of the K-Means clusters and the PCS suggested clusters were measured by the multiple class AUC, which equals to 0.991 with bootstrapped CI (0.982, 0.998). The general characteristic of the three test phenotypes and distribution of persistent symptoms was similar to the main analysis, as shown in Appendix A and Appendix A. This indicates that the results of the main analysis are quite robust. 

### 3.6. Association of the PCS Score with Change in General Health and Work Ability

The association between PCS score and the difference between the reported work ability and general health in the survey and the retrospectively assessed work ability and general health before the COVID-19 infection was assessed. There was a statistically significant association between the PCS score and the decrease in both work ability, Figure 4A and general health status (Figure 4B).

### 3.7. The Predictors of the PCS Score Groups

Twenty-two different factors were considered as possible predictors of the PCS score group severity (non/mild, moderate or severe). Through backward selection procedure to minimize the Akaike Information Criterion, 10 factors were ultimately selected: sex, BMI, smoking, snuff use, heart disease, lung disease, depression/anxiety, type 2 diabetes, chronic pain and self-reported symptom severity at onset. Of these, six factors (BMI, smoking, snuff use, diabetes mellitus type 2, chronic pain and self-reported symptom severity at onset) were found to be significant (Table 4). The Nagelkberke pseudo-R-squared of the final model was 0.44.

## 4. Discussion

The main result of our study was that we identified three distinct groups (phenotypes) of long COVID through the PCS score. These three phenotypes had a clear gradient of post-COVID symptoms’ frequency and severity and differed in sociodemographic, clinical factors and functional limitations. We further showed that higher PCS score was associated with higher reduction of self-reported general health status and working ability following COVID-19. Finally, we demonstrated that sex, BMI, smoking, snuff, coexisting heart disease, lung disease, depression/anxiety, diabetes mellitus type 2, chronic pain and self-reported severity COVID-19 onset were selected predictors of the PCS score groups.

Two previous studies used similar clustering methods and PCS score [15,16]. Bahmer et al. applied the method to the German population-based cohort of previously non-hospitalized and hospitalized individuals between 7.5 and 11.5 months following acute COVID-19 [15]. Frontera et al. based their analyses on the American prospective cohort of previously hospitalized COVID-19 patients 12 months after discharge [16]. Similar to both studies [15,16], we showed that the study subjects with the highest PCS score were characterized by higher BMI and more frequent pre-existing disease such as hypertension, heart disease and diabetes mellitus type 2.

We also found that individuals with highest PCS score (severe phenotype) had more persistent symptoms such as fatigue and memory and concentration impairment. This is in accordance with previous studies; however, it is still not known which symptoms have the most clinical relevance [9]. Furthermore, the pathophysiological mechanism that influences the course of COVID-19 sequel is not well-understood. In accordance with the hypothesis, persistent symptoms following acute COVID-19 are linked to a persistent inflammatory state that causes immunological impairment and dysfunction of various organs [25]. In our study, there was a high prevalence of self-reported diagnosed depression and anxiety in the severe PCS score group. The responders in this group also reported a high proportion of depressive and anxiety as persistent symptoms. A previous study reported that neurocognitive persistent symptoms were associated with increased odds of depression and anxiety [26].

In the present study, we found that the subjects with the severe phenotype had the highest self-reported reduction of general health status and work ability following COVID-19 and were more frequently on sick leave. Our results are in general agreement with previous studies. More severely impaired quality of life and limitations with daily living was associated with a higher persistent symptom load [15,27]. An American cluster analysis reported that half of the long COVID patients with higher severity of persistent symptoms (higher PCS) experienced disruption of work [16]. An Irish cluster study found that loss of productivity due to long COVID was more pronounced in the study subjects reporting persistent symptoms such as fatigue and pain [14].

In our study, hospitalization due to COVID-19 was not associated with severity of persistent symptoms. This in accordance with previous cluster analysis using PCS score [12,15]. On the other hand, there are also other studies reporting an increased the risk of long-term persistent symptoms in hospitalized patients [28,29,30]. The reason for these conflicting results might depend on study population and methods.

Our study identified six significant predictors of a higher PCS score: BMI, smoking, snuff, diabetes mellitus type 2, chronic pain and self-reported symptom severity at COVID-19 onset. Our finding was in general consistent with previous reports [10,12,15]. In contrast to our study, several previous reports have demonstrated that women have a higher risk of long-term symptoms following COVID-19 compared to men [10,15]. We have no obvious explanation for this discrepancy with previous studies.

This study differs from other cluster analysis [14,15,16] since it addressed the difference of general health and working ability between the period before and after COVID-19. In addition, unlike other cluster analysis [12,16,31], we also validated three phenotypes by repeated cluster analysis in randomly selected study subjects. Additionally, up to now, there has been no previous study distinguishing phenotypes of long COVID in a Swedish population.

The strength of our study it was the use of three cohorts of patients that underwent COVID-19 of various severity and over different periods of the pandemic in Sweden. The same questionnaire was used and, therefore, sociodemographic and clinical data collected in each of the populations could be combined and compared. The PCS scores for each symptom calculated in our study might be used in clinical practice in patients with long-term remaining symptoms to estimate the risk zone for severe long COVID. For example, the patients with several remaining symptoms including fatigue, memory and concentration problems, dyspnea, heart palpitation, depression, anxiety and sleep problems have PCS score sum 40.6 and may have the most severe phenotype of long COVID. These patients may need more focused individual care and rehabilitate as expected to suffer functional impairments.

A weakness of our study is that the response frequency was rather low and varied between populations. Another potential limitation was that several study subjects in the post-COVID cohort did not have a laboratory-confirmed COVID-19 infection. This was due to the limitation of laboratory testing in the early pandemic phase in Sweden. In addition, symptoms’ severity from the infection onset, self-reported general health status and work ability before the infection was assessed and rated retrospectively. Retrospective reporting on health-related quality of life may potentially introduce recall bias. Another limitation might be that the study subjects came from distinctive time periods with different availability to vaccine and various variants of virus that may change the severity and long-term outcomes of the infection.

## 5. Conclusions

The results of this study suggested three phenotypes of long COVID, where the most severe symptom load was associated with the highest impact on general health status and working ability. Knowledge on long COVID phenotypes could be used by clinicians to support their medical decisions regarding prioritizing and more detailed follow-up of some patient’s groups. Further studies are needed to determine the underlying mechanism behind these phenotypes. 

## Figures and Tables

**Figure 1 jcm-12-03617-f001:**
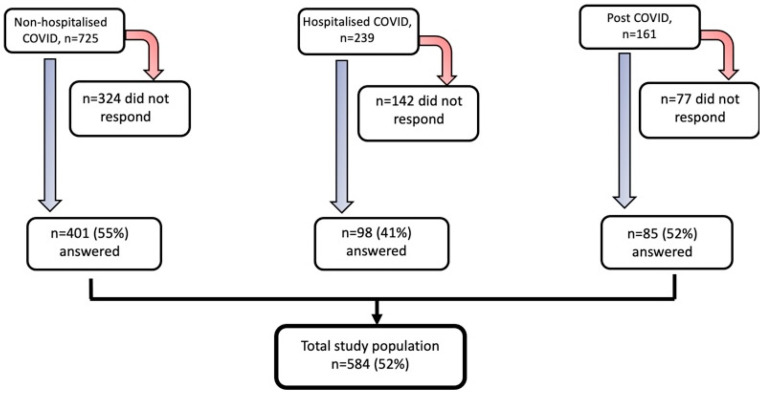
The flow chart of the study population.

**Figure 2 jcm-12-03617-f002:**
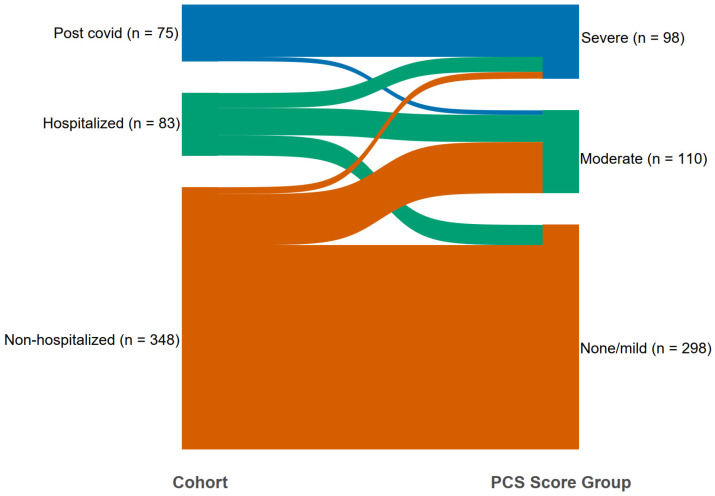
The Sankey chart for individuals from the cohorts (left) to the PCS clusters (right). In the non-hospitalized cohort, 77.9% individuals go to cluster none/mild, 19.5% of individuals go to cluster moderate (PCS cluster) and 2.6% go to cluster severe. In the hospitalized cohort, 32.5% of individuals go to cluster none/mild, 43.4% go to cluster moderate (PCS cluster) and 24.1% go to cluster severe. Lastly, in post-COVID cohort, 0% of individuals go to cluster none/mild, 8.0% go to cluster moderate and 92.0% of individuals go to cluster severe.

**Figure 3 jcm-12-03617-f003:**
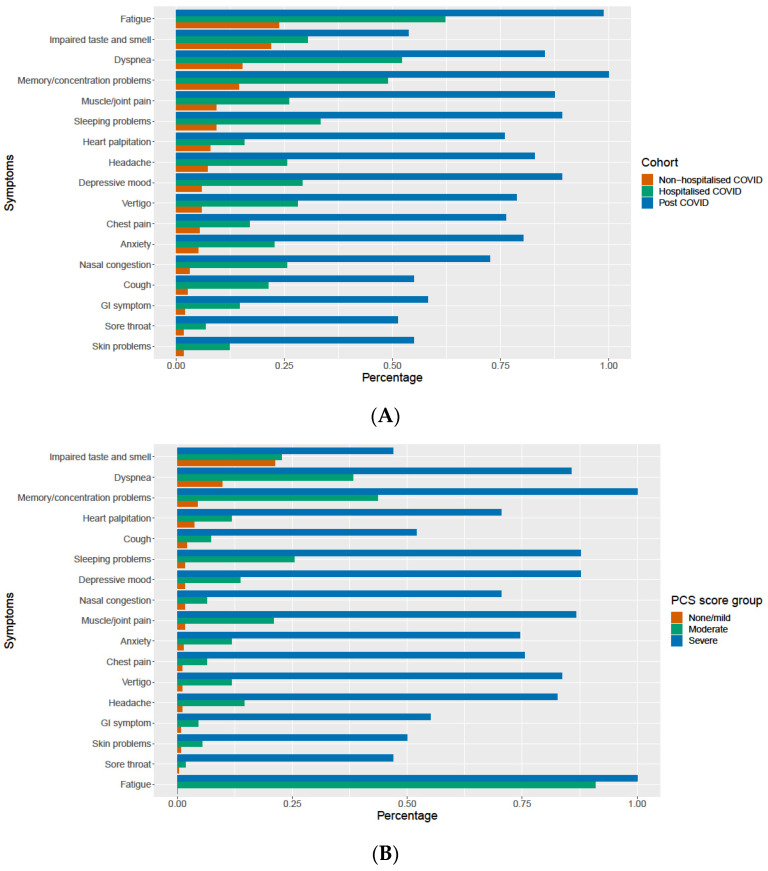
(**A**) The frequency of long-term remining symptoms in three cohorts and in (**B**) PCS score groups/phenotypes.

**Figure 4 jcm-12-03617-f004:**
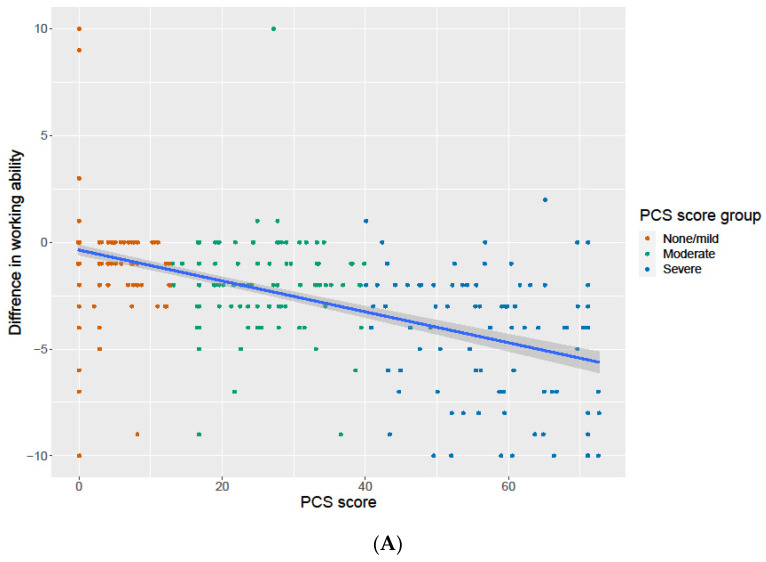
(**A**) The estimated Pearson correlation between PCS scores and the differences in working ability is −0.620 (*p* < 0.001), with a bootstrapped confidence interval of −0.691 to −0.531. The color of the dots denotes which groups they come from. (**B**) The estimated Pearson correlation between the PCS scores and the differences in health is −0.429 (*p* < 0.001), with a bootstrapped confidence interval of −0.523 to −0.326. The color of the dots denotes which groups they come from.

**Table 1 jcm-12-03617-t001:** Basic characteristic of three cohorts and disease severity during and after the acute phase of COVID-19.

	Non-Hospitalized COVID (n = 401)	Hospitalized COVID (n = 98)	Post-COVID (n = 85)	*p*-Values, ^+^
Sociodemographic and lifestyle				
Age, mean (SD)	44.2 (13.8)	58.5 (10.1)	50.0 (11.9)	**<0.001**
Female, n (%)	298 (74.3)	43 (44.3)	55 (64.7)	**<0.001**
Country of birth, n (%) Sweden	319 (79.8)	74 (78.7)	74 (87.1)	0.263
Education level, n (%) Up to secondary school Vocational education University	141 (35.3) 37 (9.3) 221 (55.4)	50 (51.5) 18 (18.6) 29 (29.9)	35 (42.7) 13 (15.9) 34 (41.4)	**<0.001**
Working status, n (%) Working Parental leave Looking for a job Retired Sick leave Student	350 (87.3) 5 (1.2) 2 (0.5) 11 (2.7) 18 (4.5) 6 (1.5)	70 (71.4) 0 (0) 2 (2.0) 18 (18.4) 5 (5.1) 1 (1.0)	52 (61.9) 0 (0) 2 (2.4) 5 (6.0) 22 (26.2) 3 (3.5)	**<0.001**
Marital status, n (%) Married Partner Divorced or separated Widower/er Single	134 (33.5) 139 (34.7) 75 (18.7) 15 (3.8) 37 (9.3)	58 (59.8) 18 (18.6) 14 (14.4) 3 (3.1) 4 (4.1)	41 (48.2) 21(24.7) 6 (7.1) 0 (0) 17 (20.0)	**<0.001**
Smoking, n (%) Never smoked Ex-smoker Current smoker	303 (76.1) 82 (20.6) 13 (3.3)	55 (58.5) 36 (38.3) 3 (3.2)	2 (2.4) 28 (32.9) 55 (64.7)	**<0.001**
Snuff, n (%)	316 (81.0)	19 (20.4)	13 (15.5)	**<0.001**
Pre-existing comorbidities, n (%) and BMI				
BMI, mean (SD)	25.7(4.7)	30.1 (6.3)	28.4 (6.1)	**<0.001**
Hypertension	50 (13.7)	41 (42.3)	27 (31.8)	**<0.001**
Heart disease	14 (3.9)	7 (7.2)	6 (7.1)	0.265
Hypo/hyperthyroidism	33 (9.1)	4 (4.1)	6 (7.1)	0.245
Diabetes	14 (3.9)	12 (12.2)	4 (4.7)	**0.006**
Lung disease	45 (12.4)	25 (25.5)	24 (28.2)	**<0.001**
Liver disease	1 (0.3)	0 (0.0)	1 (1.2)	0.382
Cancer	21 (5.8)	11 (11.3)	3 (3.5)	0.078
Immunosuppressive treatment	16 (4.5)	4 (4.1)	4 (4.7)	0.971
Depression/Anxiety	93 (25.3)	22 (22.4)	34 (40.0)	**0.012**
Chronic pain	18 (5.0)	9 (9.3)	23 (27.1)	**<0.001**
Other measurements				
Symptom severity at onset, median (IQR)	3 (2, 3)	4 (3, 4)	4 (3, 4)	**<0.001**
Hospitalized, n (%)	0	98 (100)	21 (25.0)	**<0.001**
Laboratory-confirmed COVID-19 infection n (%)	401 (100)	98 (100)	56 (67.5)	**<0.001**
Number of months from infection onset, median	12	12	22 (IQR:18, 27)	**<0.001**
Mean number of remaining symptoms, mean (SD)	1.3 (2.1)	4.3 (4.5)	12.3 (3.7)	**<0.001**
Health status COVID-19, median (IQR)	90 (85, 95)	90 (10, 95)	Missing variable	**0.045**
Health status today, Median (IQR)	80 (70, 90)	70 (55, 85)	40 (20, 60)	**<0.001**
Difference of health status COVID-19 and today, Median (IQR)	−5 (−15, 0)	−10 (−25, −5)	Missing variable	**<0.001**
Working ability COVID-19, median (IQR)	10 (9, 10)	10 (7, 10)	10 (9, 10)	0.306
Work ability today, Median (IQR)	9 (8, 10)	8 (4, 9)	4 (1, 6)	**<0.001**
Difference working ability COVID-19 and today, median (IQR)	0 (−1.3, 0)	−1 (−2, 0)	−5 (−8, −3)	**<0.001**

+, *p* values shown in bold are significant under the level of 0.05.

**Table 2 jcm-12-03617-t002:** The 17 long-term symptoms were arranged in order of the weight assigned to each indicator in the PCS score, starting with the highest weight.

No	Symptom Complex	Cluster I Center (n = 299)	Cluster II Center (n = 120)	Cluster III Center (n = 87)	Regression Coefficient	PCS * Score Weight
2	Fatigue	0	0.933	0.989	16.758	16.8
15	Memory and concentration problems	0.043	0.492	1.000	8.144	8.1
4	Sore throat	0.003	0.017	0.529	7.425	7.4
3	Muscles and joints pain	0.020	0.217	0.931	5.849	5.8
1	Cough	0.020	0.075	0.575	4.706	4.7
11	Heart palpitation	0.037	0.133	0.759	4.500	4.5
7	Vertigo	0.013	0.15	0.874	4.031	4.0
6	Headache	0.010	0.192	0.851	4.013	4.0
14	Depressive mood	0.017	0.175	0.920	4.003	4.0
10	Chest pain	0.013	0.083	0.805	3.127	3.1
12	GI symptoms	0.007	0.05	0.609	2.934	2.9
5	Dyspnea	0.097	0.408	0.885	2.825	2.9
16	Sleep problems	0.017	0.308	0.885	2.250	2.3
13	Anxiety mood	0.013	0.15	0.782	2.086	2
17	Impaired taste and smell	0.214	0.208	0.517	0.046	0
9	Nasal symptoms	0.017	0.083	0.759	−0.150	0
8	Skin problems	0.007	0.058	0.552	−1.448	−1.5

* Post-COVID syndrome (PCS) score.

**Table 3 jcm-12-03617-t003:** Characteristic of three clusters distinguished by PCS score. *p*-value was adjusted for control the false discovery rate within three groups: sociodemographic and lifestyle, pre-existing comorbidities and BMI and other measurements (n = 506).

Characteristics	None/Mild PCS * Score ≤ 13 (n = 298)	Moderate PCS * Score > 13 and ≤40 (n = 110)	Severe PCS * Score > 40 (n = 98)	*p*-Value Unadjusted, ^+^	*p*-Value Adjusted, ^+^
Sociodemographic and lifestyle					
Age, mean (SD)	44.2 (13.9)	50.3 (13.6)	49.8 (12.2)	**<0.001**	**0.002**
Female, n (%)	211 (71.0)	71 (64.5)	65 (66.3)	0.387	0.397
Country of birth n (%) Sweden	246 (84.0)	86 (78.2)	81 (82.7)	0.397	0.397
Education level, n (%) Up to Gymnasium Two years Three years	102 (34.5) 27 (9.1) 167 (56.4)	45 (41.3) 14 (12.8) 50 (45.9)	45 (46.4) 15 (15.5) 37 (38.1)	**0.022**	**0.028**
Working status, n (%) Working Parental leave Looking for a job Retired Sick leave Student	266 (91.1) 4 (1.4) 2 (0.7) 10 (3.4) 8 (2.7) 2 (0.7)	91 (85.8) 0 (0) 1 (0.9) 6 (5.7) 6 (5.7) 2 (1.9)	61 (63.6) 0 (0) 1 (1.0) 10 (10.4) 21 (21.9) 3 (3.1)	**<0.001**	**<0.001**
Marital status, n (%) Married Sambo Divorced Widower Single	95 (32.1) 110 (37.2) 57 (19.3) 14 (4.7) 20 (6.7)	53 (48.2) 30 (27.3) 18 (16.3) 2 (1.8) 7 (6.4)	42 (42.9) 25 (25.5) 10 (10.2) 1 (1.0) 20 (20.4)	**<0.001**	**0.002**
Smoking, n (%) Never smoked Ex-smoker Current smoker	224 (76.5) 58 (19.8) 11 (3.7)	61 (56.5) 40 (37.0) 7 (6.5)	20 (20.4) 32 (32.7) 46 (46.9)	**<0.001**	**0.002**
Snuff, n (%)	218 (75.2)	66 (61.1)	20 (20.8)	**<0.001**	**0.002**
Pre-existing comorbidities, n (%) and BMI					
BMI, mean (SD)	25.5 (4.2)	27.9 (5.4)	29.2 (7.1)	**<0.001**	**0.002**
Hypertension	39 (13.1)	24 (21.8)	32 (32.7)	**<0.001**	**0.002**
Heart disease	7 (2.3)	6 (5.5)	8 (8.2)	**0.032**	0.051
Hypo/hyperthyroidism	19 (6.4)	11 (10.6)	6 (6.1)	0.411	0.452
Diabetes	7 (2.3)	8 (7.7)	8 (8.2)	**0.017**	**0.031**
Lung disease	29 (9.7)	19 (18.6)	31 (31.6)	**<0.001**	**0.002**
Liver disease	1 (0.3)	0 (0)	1 (1.0)	0.489	0.538
Cancer	17 (5.7)	8 (7.7)	3 (3.1)	0.407	0.452
Immunosuppressive treatment	8 (2.7)	8 (7.7)	6 (6.2)	0.081	0.112
Depression/Anxiety	59 (19.8)	28 (25.4)	39 (40.2)	**<0.001**	**0.002**
Chronic pain	9 (3.0)	10 (9.2)	24 (24.5)	**<0.001**	**0.002**
Other measurements					
Symptom severity at COVID-19 onset, median (IQR)	3 (2, 3)	3 (3, 4)	4 (3, 4)	**<0.001**	**<0.001**
Hospitalized, n (%)	27 (9.1)	37 (33.6)	35 (36.5)	**<0.001**	**<0.001**
Laboratory-confirmed COVID-19, n (%)	298 (100)	109 (99.1)	71 (74.0)	**<0.001**	**<0.001**
Number of months from infection onset, mean (SD)	12.0 (0.0)	12.6 (2.5)	19.3 (6.7)	**<0.001**	**<0.001**
Mean number of remaining symptoms, mean (SD)	0.5 (0.9)	3.4 (1.8)	12.5 (3.0)	**<0.001**	**<0.001**
Health status COVID-19, median, (IQR)	90 (85, 100)	90 (85, 100)	85 (70, 95)	**0.029**	**0.033**
Health status today, median (IQR)	85 (75, 95)	70 (55, 80)	45 (30, 60)	**<0.001**	**<0.001**
Difference health status COVID-19 and today, median (IQR)	0 (10, 0)	−20 (−26.3, −10)	−20 (−35, −10)	**<0.001**	**<0.001**
Work ability COVID-19, median (IQR)	10 (9, 10)	10 (9, 10)	10 (9, 10)	0.401	0.401
Work ability today, median (IQR)	9 (8, 10)	8 (6, 9)	4 (1, 7)	**<0.001**	**<0.001**
Difference working ability COVID-19 and today, median (IQR)	0 (−1, 0)	−2 (−3, −1)	−4 (−7, −2)	**<0.001**	**<0.001**

* Post-COVID syndrome (PCS) score; +, *p* values shown in bold are significant under level 0.05.

**Table 4 jcm-12-03617-t004:** Factors predicting the post-COVID score (PCS) clusters. Missing values assessed by imputation method.

Predictor Variable	Level	Regression Coefficient			Odds Ratio		*p*-Value, ^+^
		Estimate	SD	95% CI	Estimate	95% CI	
Sex	Female	0.356	0.226	(−0.083, 0.805)	1.427	(0.920, 2.237)	0.116
BMI	Scale	0.060	0.020	(0.021, 0.099)	1.062	(1.022, 1.104)	**0.003**
Smoking	Have smoked	1.258	0.216	(0.837, 1.684)	3.519	(2.309, 5.387)	**<0.001**
Snuff	Yes	−1.108	0.219	(−1.540, −0.679)	0.330	(0.214 0.507)	**<0.001**
Heart disease	Yes	0.910	0.469	(−0.013, 1.840)	2.484	(0.987, 6.294)	0.052
Lung disease	Yes	0.468	0.273	(−0.071, 1.002)	1.596	(0.932, 2.725)	0.087
Depression/anxiety	Yes	0.408	0.234	(−0.054, 0.865)	1.503	(0.947, 2.375)	0.081
Diabetes	Yes	0.972	0.464	(0.062, 1.899)	2.644	(1.064, 6.612)	**0.036**
Chronic pain	Yes	0.722	0.360	(0.022, 1.437)	2.059	(1.022, 4.207)	**0.045**
Symptom severity at onset	1–5 scale	0.690	0.110	(0.478, 0.911)	1.995	(1.612, 2.487)	**<0.001**

+, *p* values shown in bold are significant under level 0.05.

## Data Availability

Data can be available on demand.

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
