# Peer review of "Clustering Analysis Identified Three Long COVID Phenotypes and Their Association with General Health Status and Working Ability"

_jcm, 2023, doi:10.3390/jcm12113617_

Round 1
Reviewer 1 Report
1. Please change “Severe acute respiratory syndrome coronavirus-2” to “severe acute respiratory syndrome coronavirus-2”.
2. Please change “Coronavirus disease 2019” to “coronavirus disease 2019”.
3. In the first citation, the abbreviation for the word should be preceded by the full name, and then write the abbreviation (e.g., BMI). Please correct it.
4. Please include WHO link instead of reference 2.
5. Please enter your questionnaire as a supplementary file.
6. Please explain the meaning of PCS below the tables.
7. Please include this sentence below tables.
“P value are shown in bold are significant.”
8. Please change “variable” to “symptoms” in figure 3.
9. For better understanding, please include the meaning of cluster I, II, and III below the supplementary figures.
Minor editing of English language required.
Author Response
- Please change “Severe acute respiratory syndrome coronavirus-2” to “severe acute respiratory syndrome coronavirus-2”.
Reply: Thank you for this comment. We changed in Line 39.
- Please change “Coronavirus disease 2019” to “coronavirus disease 2019”.
Reply: Thank you for this suggestion. We changed in Line 40.
- In the first citation, the abbreviation for the word should be preceded by the full name, and then write the abbreviation (e.g., BMI). Please correct it.
Reply: Thank you for this comment. We changed as suggested (BMI abbreviation), Line 29. We also added the full name of ANOVA (Page 4, Line 154) and ROC (Page 4, Line 166) when first used.
- Please include WHO link instead of reference 2.
Reply: Thank you for the comment. We replaced previous reference 2 with WHO link as suggested.
- Please enter your questionnaire as a supplementary file.
Reply: Thank you for these comments. We add the questionnaire as Appendix 1.
- Please explain the meaning of PCS below the tables.
Reply: Thank you for this comment. We explained PCS below the Tables 2 and 3.
- Please include this sentence below tables.“P value are shown in bold are significant.”
Reply: Thank you for this comment. We changed as suggested.
- Please change “variable” to “symptoms” in figure 3.
Reply: Thank you for this comment. We changed as suggested.
- For better understanding, please include the meaning of cluster I, II, and III below the supplementary figures.
Reply: Thank you for this comment. We changed as suggested.
Additionally, we made some extra changes in order to improve the manuscript:
- We put the description of Figure 1-4 under the figures.
- We adjusted Table 1-4 to make them look better, i.e., remove the horizontal lines, and put shadows instead.
- We improved the quality of Figure 2, Figure 3, and Figure 4.
- We deleted one redundant row (Symptoms severity at COVID onset) in Table 3.
- We did some minor corrections in Table 1.
- Some other minor format corrections to keep the whole paper consistent.
- The manuscript underwent minor English revision.
Reviewer 2 Report
Clustering analysis – Journal of Clinical Medicine
Very interesting study. The main result of the study was the identification of three distinct groups or phenotypes using post-COVID syndrome score. A clear gradient has been found in post COVID symptoms frequency and severity with difference in sociodemographic, clinical factors, and functional limitations. It has been shown that higher PCS score was associated with higher reduction of self-reported general health status and working ability following COVID-19.
Introduction
Line 65 – “Few previous studies of long COVID clusters have addressed work ability” – there are no citations after this sentence. References should be added and main findings described in short.
Table 1 – Please check – Marital status of Post COVID group – does not sum up in 100 %.
Discussion
I suggest that you add to Discussion section what was new and original in your study compared to previous published studies.
Reference
Line 425 Reference No. 6 – Add missing information to the reference.
Author Response
Introduction
Line 65 – “Few previous studies of long COVID clusters have addressed work ability” – there are no citations after this sentence. References should be added and main findings described in short.
Reply: Thank you for this comment. We changed as suggested, Line 66-68.
Table 1 – Please check – Marital status of Post COVID group – does not sum up in 100 %.
Reply: Thank you for this comment. We corrected this, Table 1.
Discussion
I suggest that you add to Discussion section what was new and original in your study compared to previous published studies. Reply: It is an important comment. We completed the discussion, line 345-350.
Reference
Line 425 Reference No. 6 – Add missing information to the reference.
Reply: Thank you for this comment. We add missing information to thee reference 6.
Additionally, we made some extra changes in order to improve the manuscript:
- We put the description of Figure 1-4 under the figures.
- We adjusted Table 1-4 to make them look better, i.e., remove the horizontal lines, and put shadows instead.
- We improved the quality of Figure 2, Figure 3 and Figure 4.
- We deleted one redundant row (Symptoms severity at COVID onset) in Table 3.
- We did some minor corrections in Table 1.
- Some other minor format corrections to keep the whole paper consistent.
- The manuscript underwent minor English revision.